# The Importance of mTORC1-Autophagy Axis for Skeletal Muscle Diseases

**DOI:** 10.3390/ijms24010297

**Published:** 2022-12-24

**Authors:** Xujun Han, Kah Yong Goh, Wen Xing Lee, Sze Mun Choy, Hong-Wen Tang

**Affiliations:** 1Program in Cancer and Stem Cell Biology, Duke-NUS Medical School, 8 College Road, Singapore 169857, Singapore; 2Division of Cellular & Molecular Research, Humphrey Oei Institute of Cancer Research, National Cancer Centre Singapore, Singapore 169610, Singapore

**Keywords:** mTORC1, autophagy, muscle diseases

## Abstract

The mechanistic target of rapamycin (mTOR) complex 1, mTORC1, integrates nutrient and growth factor signals with cellular responses and plays critical roles in regulating cell growth, proliferation, and lifespan. mTORC1 signaling has been reported as a central regulator of autophagy by modulating almost all aspects of the autophagic process, including initiation, expansion, and termination. An increasing number of studies suggest that mTORC1 and autophagy are critical for the physiological function of skeletal muscle and are involved in diverse muscle diseases. Here, we review recent insights into the essential roles of mTORC1 and autophagy in skeletal muscles and their implications in human muscle diseases. Multiple inhibitors targeting mTORC1 or autophagy have already been clinically approved, while others are under development. These chemical modulators that target the mTORC1/autophagy pathways represent promising potentials to cure muscle diseases.

## 1. Introduction of mTORC1

The mechanistic target of rapamycin (mTOR) is an evolutionarily conserved serine/threonine kinase. It was first identified as a target protein of rapamycin through the screening of rapamycin-resistant mutants in yeast [1]. Rapamycin can form a complex with FK506-binding protein (FKBP) that inhibits cell cycle progression by inhibition of the yeast TOR1 and TOR2 [1]. Subsequent studies in mammalian cells identified the homologous protein, termed mTOR, which shares more than 40% conservation in amino acid sequence with yeast TOR1 and TOR2 [2]. mTOR exists as two structurally and functionally different complexes, known as mTOR complex 1 (mTORC1) and 2 (mTORC2) respectively (Figure 1). mTORC1 is composed of the central mTOR kinase, the scaffolding protein LST8 [3], the inhibitory subunit DEPTOR [4], the Tti1/Tel2 complex required for mTOR complex stability and assembly [5], Raptor, and PRAS40. Raptor (regulatory-associated protein of mTOR) is critical for mTORC1 assembly, stability, and substrate specificity [6,7], and associates with PRAS40 (Proline-rich AKT substrate 40 kDa), which functions as an inhibitor of mTORC1 activity [8]. mTORC2 shares the components mTOR, LST8, DEPTOR and Tti1/Tel2 that are similarly present in mTORC1, in addition to the unique subunits, Rictor and SIN1 [9]. mTORC1 and mTORC2 can be distinguished by their acute response to rapamycin [10]. Rapamycin is an antifungal metabolite produced by *Streptomyces hygroscopicus,* which possess immunosuppressive and anti-proliferative properties in mammalian cells. Rapamycin allosterically inhibits mTORC1. In contrast, mTORC2 shows insensitivity to acute rapamycin treatment [10,11].

Genetic studies and pharmacological research using rapamycin have demonstrated that mTORC1 is a master regulator of cell growth and metabolism which senses and integrates different nutritional and environmental cues, including growth factors, energy levels, cellular stress, and amino acids [12]. Growth factor activation of mTORC1 is mediated by a Ras family GTPase, RAS homolog enriched in brain (Rheb), following receptor-coupled PI3K activation and the AKT-dependent phosphorylation and inhibition of the tuberous sclerosis complex (TSC) 1/2 [13]. The TSC complex can also be stabilized by LKB1-AMPK signaling [14]. In contrast, amino acids can stimulate mTORC1 in a PI3K/Akt axis-independent manner and induce the translocation of mTORC1 to the lysosomal surface where mTORC1 is activated upon contact with Rheb. This process requires many complexes, such as the v-ATPase, Ragulator, the Rag GTPases, and GATOR1/2 [15].

## 2. Regulation of Autophagy by mTORC1

In the absence of nutrients or growth factors, mTORC1 activity is reduced. In turn, it activates cellular catabolic processes while suppressing anabolic processes. Reduced mTORC1 activity increases the activity of eukaryotic translation initiation factor 4E-binding protein 1 (4E-BP1) and inhibits ribosomal protein S6 kinase 1 (S6K1) to block protein translation [16]. mTORC1 inactivation also activates autophagy, an evolutionarily conserved catabolic process that recycles long-lived proteins or damaged organelles and provides energy or macromolecular precursors for cell survival [17,18]. This degradation process is coordinately modulated by multiple autophagic regulators [19,20,21,22,23]. Among these regulators, mTORC1 signaling plays critical roles in regulating each step of the autophagy process, including induction, nucleation, elongation, and formation of a double-membrane autophagosome, followed by the fusion of the autophagosome with a lysosome to form autolysosomes to degrade and recycle autophagosome-sequestered substrates (Figure 2). Here, we review the critical functions of mTORC1 in regulating autophagy.

### 2.1. Regulation of ULK1 Complex during Autophagy Initiation

In mammals, the activation of Unc51-like kinase 1 (ULK1) complex, composed of the Ser/Thr kinase ULK1, ATG13, FIP200, and ATG101, is essential for autophagy induction. The ULK1 complex serves as the main interface where the direct intervention of mTORC1 in autophagy occurs and its activity is mainly regulated by mTORC1- and AMP-activated protein kinase (AMPK)-mediated phosphorylation [24,25]. In nutrient-rich conditions where mTORC1 is active, mTORC1 phosphorylates ULK1 at the Ser757 site, disrupting the interaction of AMPK and ULK1, leading to the inhibition of autophagy induction [24]. mTORC1 also phosphorylates ATG13, a component of ULK1 complex, at Ser258 to decrease ULK1 complex activity and suppress autophagy [26]. During starvation and environmental stresses where mTORC1 is inactive, mTORC1 dissociates from the ULK1 complex and ULK1 is activated by AMPK-dependent phosphorylation at multiple sites [24]. ULK1 phosphorylation by AMPK is able to induce autophagy [24]. In addition, recent studies suggest that mTORC1-mediated autophagy via ULK1 can also be regulated by protein kinase G 1 (PKG1) in the heart [27,28]. Together, these findings suggest that mTORC1 plays critical roles in autophagy induction through the regulation of the ULK1 complex.

### 2.2. Regulation of PI3KC3-Beclin 1-ATG14 Complex during Nucleation

After ULK1-mediated autophagy initiation, the nucleation of phagophore requires the lipid kinase activity of class III phosphatidylinositol 3-kinase (PI3KC3), also known as Vacuolar protein sorting 34 (Vps34), which generates phosphatidylinositol-3 phosphate (PI3P) from phosphatidylinositol at the phagophore to promote pre-autophagosome formation. PI3KC3 forms two distinct complexes (PI3KC3-CI and PI3KC3-CII). PI3KC3, VPS15, and Beclin 1 make up the core subunits that are conserved in both complexes. Complex I (PI3KC3-CI) contains ATG14, activating molecule in Beclin 1-regulated autophagy protein 1 (AMBRA1) and nuclear receptor-binding factor 2 (NRBF2), while complex II includes UV radiation resistance-associated gene protein (UVRAG) [29]. Among them, PI3KC3-CI has been shown to regulate the nucleation of phagophores. Upon mTORC1 inhibition or ULK1 complex activation, ULK1 complex targets the sub-domains of the endoplasmic reticulum (ER), known as omegasomes, and recruits PI3KC3-CI to produce PI3P at phagophore, facilitating the nucleation of autophagosomes [30,31].

mTORC1 inhibition activates the ULK1 complex, which in turn induces the phosphorylation of Beclin 1 at Ser15 and Ser30, thus causing the activation of the PI3KC3-CI [32,33]. In addition, it has been reported that mTORC1 can also directly regulate the activity of the PI3KC3-CI by phosphorylating its components ATG14, AMBRA1, and NRBF2. mTORC1 inhibits PI3KC3 activity by directly phosphorylating ATG14 on multiple sites. Mutation of these phosphorylation sites on ATG14 enhances autophagic flux [34]. NRBF2 can be phosphorylated by mTORC1 at S113 and S120. The inhibition of such phosphorylation events increases VPS34 complex assembly and activity, enhancing autophagy flux [35]. In addition, mTORC1 can directly phosphorylate AMBRA1 at Ser52 under normal conditions. The inhibition of mTORC1 results in the dephosphorylation of AMBRA1 and increases its interaction with E3-ligase TRAF6, which subsequently ubiquitinates ULK1 on Lys-63. This K63-linked ubiquitination stabilizes ULK1 and induces its kinase activity, leading to an increase in autophagy induction [30]. Thus, these studies suggest that mTORC1 strictly restricts the nucleation of phagophores through the targeting of multiple components of the PI3KC3-CI complex.

### 2.3. Regulation of Autophagosome Expansion

When PI3P is generated at the omegasome, PI3P subsequently recruits PtdIns3P-binding effectors, including Atg18/WD-repeat domain phosphoinositide-interacting protein-2 (WIPI-2) and FYVE domain-containing protein 1 (DFCP1), which in turn recruit more autophagy machinery proteins [36,37]. WIPI2 binds to ATG16L and recruits the ATG12-ATG5-ATG16L complex to the phagophore. The WIPI-2-recruited ATG12-ATG5-ATG16L complex is part of a conjugation system where the E1-like ATG7 transfers LC3-I to the E2-like enzyme ATG3 that associates with ATG12 on the complex, catalyzing the conjugation of ATG8/LC3 proteins with membrane resident phosphatidylethanolamine (PE) [38]. This lipidation of the ATG8/LC3 process generates LC3-II, which is the characteristic signature of autophagic membranes and is involved in the ATG9 vesicle sequestration of cargo [39,40].

Multiple studies have demonstrated that mTORC1 negatively regulates autophagosome expansion by phosphorylating WIPI2 and p300 acetyltransferase [41,42]. mTORC1 phosphorylates WIPI2 at Ser395 and the phosphorylated WIPI2 interacts with E3 ligase HUWE1 to promote its proteasomal degradation. The mTORC1-regulated protein stability of WIPI2 affects both basal and starvation-induced autophagy [41]. Moreover, mTORC1 also directly phosphorylates p300 acetyltransferase to activate p300 by relieving its autoinhibition. P300-mediated LC3 acetylation prevents its interaction with E1 ubiquitin ligase ATG7, thus leading to autophagy inhibition. Upon starvation, reduced mTORC1 activity induces the dephosphorylation of p300. This dephosphorylation event inactivates p300 and causes the deacetylation of LC3, thereby increasing LC3-ATG7 interaction, LC3 lipidation, and autophagosome expansion [42].

### 2.4. Regulation of Autophagosome Maturation and Termination

After becoming fully sealed, the autophagosome subsequently fuses with lysosome to degrade the engulfed material. Autophagosomes and lysosomes are tethered by diverse tethering machinery, such as Rab GTPases, homotypic fusion and vacuole protein sorting (HOPS) complex, ATG14, and UVRAG [43,44,45]. The fusion of the autophagosome membrane with the lysosome membrane involves the HOPS complex that tethers autophagosomes to lysosomes as well as Syntaxin-17 (Stx-17) to facilitate membrane fusion [46]. Pacer was identified as a vertebrate-specific autophagy activator and is found to facilitate the biogenesis of PI3P on autophagosomes. Furthermore, Pacer interacts with Stx17 and recruits HOPS to autophagosomes [47]. In addition, UVRAG, a component of the PI3KC3 complex II (PI3KC3-CII), is required for the interaction between the autophagosome and PI3KC3-CII. The association of UVRAG with HOPS increases autophagosome-lysosome fusion [44]. At the last stage of autophagy, lysosomal membranes are recycled from autolysosomes to maintain lysosome homeostasis, a process termed Autophagic lysosome reformation (ALR) [48].

It has been shown that mTORC1 directly interacts with and phosphorylates UVRAG at Ser498 in nutrient-rich conditions [49]. This mTORC1-dependent phosphorylation of UVRAG inhibits the interaction between UVRAG and the HOPS complex. However, upon starvation, the dephosphorylation of UVRAG induced by mTORC1 inactivation promotes its interaction with HOPS and enhances autophagosome maturation [49]. Besides Ser498, mTORC1 also phosphorylates UVRAG at Ser550 and Ser571 to maintain ALR [50]. The phosphorylation of UVRAG at Ser550 and Ser571 induces the activation of PI3KC3 and generates PI3P at lysosome. The mutation of these mTORC1-dependent phosphorylation sites on UVRAG causes the failure of lysosome regeneration and induces cell death during starvation, suggesting the indispensable function of mTOR in ALR [50]. A recent study has identified Pacer as a substrate of mTORC1. mTORC1 phosphorylates Pacer at Ser157 and this phosphorylation abolishes the interaction of Pacer with HOPS and Stx17, thus preventing autophagosome maturation during nutrient-rich conditions [51]. During starvation, dephosphorylated Pacer in turn recruits HOPS complex for autophagosome maturation [51].

### 2.5. mTORC1-Dependent RNA Metabolism

While earlier studies mostly focused on mTORC1 downstream signals linked to cytoplasmic protein metabolism, recent studies have shown that mTORC1 signaling plays critical roles in RNA metabolism, ranging from pre-mRNA splicing, polyadenylation, and mRNA methylation. mTORC1 activation has been shown to induce global mRNA 3′-UTR shortening [52]. In our previous study, we further found that mTORC1 regulates RNA processing of autophagy-related gene (Atg) transcripts and alters ATG protein levels and activities through the cleavage and polyadenylation (CPA) complex. Specifically, mTORC1 activity suppresses CDK8 and DOA/CLK2 kinases, which directly phosphorylate CPSF6, a component of the CPA complex. The phosphorylation status of CPSF6 affects its localization, RNA binding, and starvation-induced alternative RNA processing of Atg1/ULK1 and Atg8/LC3 transcripts, nutrient, and energy metabolism, thereby modulating autophagy and metabolism [53]. Alternatively, mTORC1 also regulates the phosphorylation of the decapping enzyme Dcp2. Phosphorylated Dcp2 associates with RCK family members and binds to Atg8/LC3 transcripts to degrade them, leading to autophagy inhibition [54]. Furthermore, recent studies further demonstrate the roles of mTORC1 in RNA methylation. mTORC1-activated S6K enhances the translation of Wilms’ tumor 1-associated protein (WTAP), an adaptor for the *N*6-methyladenosine (m^6^A) RNA methyltransferase complex, leading to an increase in m^6^A levels. Increased m^6^A activity promotes c-Myc transcriptional activity and the proliferation of mTORC1-activated cancer cells [55]. In addition, we found that mTORC1 stabilizes the m^6^A methyltransferase complex through the chaperonin containing TCP1 (CCT) complex. The upregulation of m^6^A modification promotes the degradation of Atg transcripts, including Atg1 and Atg8, and inhibits autophagy [56]. Collectively, these studies uncover another layer of mTORC1 regulation of autophagy through RNA metabolisms.

## 3. Roles of the mTORC1-Autophagy Pathway in Regulating Skeletal Muscle Functions

Skeletal muscle is the most abundant tissue, comprising ~40% of body mass in humans, and plays key roles in locomotion and maintaining metabolism. It serves as a protein reservoir in the human body which undergoes rapid turnover, a process strictly controlled by the balance between protein synthesis and degradation. As a result, skeletal muscle possesses an extreme sensitivity to the changes in both autophagy and mTORC1 activities. To date, excessive muscle loss has been used as a prognostic index of negative outcomes for a variety of diseases ranging from cancer, infections, and unhealthy aging [57].

### 3.1. mTORC1, but Not mTORC2, Regulates Skeletal Muscle Sizes

In response to exercise or hormonal stimulation, new proteins are generated, increasing cellular volume and muscle growth, a process named hypertrophy. In contrast, catabolic conditions such as cancer, infections, diabetes, aging, or inactivity/disuse promote a net loss of proteins, causing shrinkage of the muscle volume, a condition named atrophy [58]. Therefore, the balance between biogenesis versus destruction defines the size and the function of muscle cells.

Altered mTOR activity has been linked to both muscle hypertrophy and atrophy. Muscle-specific *mTOR* knockout mice exhibit severe myopathy leading to premature death between 22 and 38 weeks of age [59]. The *mTOR* and *Raptor* (the scaffold protein of mTORC1) knockout mice exhibited decreased postnatal growth due to the reduced size of fast-twitch muscles and displayed a progressive muscle atrophy phenotype [59,60]. However, muscle-specific *Rictor* (a component of mTORC2) knockout mice fail to show any significant phenotype. Similarly, a recent study reported that muscle specific *mTOR* and *Raptor* double knockout in mice induces muscle atrophy and a slower muscle relaxation, which may be caused by a shift of muscle types, from fast-twitch fibers to slow-twitch fibers, and changes in the expression levels of calcium-related genes. The double knockout mice exhibit more severe phenotypes compared to the mice with the deletion of either *Raptor* or *mTOR* alone [61]. Consistent with these results, the treatment of rapamycin, an inhibitor of mTORC1, suppresses muscle growth [62]. The deletion of muscular S6K1, a mTORC1 downstream target, induces energy stress and muscle cell atrophy [63]. These studies demonstrate that mTORC1, not mTORC2, is the major regulator in the control of muscle fiber size.

### 3.2. The Roles of mTORC1-Autophagy Axis in Muscle Homeostasis

Interestingly, an early study reported that acute activation of mTORC1 in vivo drives muscle hypertrophy in the short-term [64]; however, chronic mTORC1 activation by TSC1 depletion in the muscle leads to severe and progressive muscle atrophy, along with low body mass and early death [65]. A recent study also observed that activation of mTORC1 in aged muscle leads to progressive muscle fiber damage, fiber death, and loss of muscle mass [66]. These muscle atrophy phenotypes induced by mTORC1 hyperactivation are primarily due to the lack of autophagy in muscles [65]. Autophagy provides energy and building blocks for metabolisms and thus regulates the level of amino acids, lipids, carbohydrates, and nucleic acids [67,68]. It is also required for intracellular quality control. The inhibition of autophagy leads to the accumulation of ubiquitinated protein aggregates and inclusion bodies as well as cause abnormalities in mitochondria, peroxisomes, ER, and Golgi. For instance, similar to the phenotypes in *TSC1*-deficient mice, the muscle-specific deletion of *Atg7* causes muscle atrophy and muscle force decreases [69]. The accumulation of abnormal mitochondria, sarcoplasmic reticulum distension, the disorganization of sarcomere, and the formation of aberrant concentric membranous structures were observed in *Atg7*-null muscles, showing that autophagy is essential to preserving muscle mass and maintaining muscle integrity [69]. Therefore, these results suggest that both chronically aberrant increases and decreases in mTORC1 activity and deregulated autophagy result in muscle atrophy, implying that dynamic synthesis–degradation oscillations modulated by the balance of the mTORC1-autophagy axis are essential for maintaining muscle homeostasis (Figure 3).

### 3.3. The Roles of FOXO and mTORC1 in Regulation of Muscle Autophagy

The forkhead box O (FOXO) family of transcription factors are downstream targets of Akt [70]. The Akt-dependent phosphorylation of FOXOs results in nuclear exclusion, thus impairing the functions of FOXOs. In skeletal muscle, the activation of FOXO1 and 3 have been linked to multiple muscle atrophy conditions such as sarcopenia and cachexia [71,72]. When Akt is inactive, dephosphorylated FOXOs translocate from the cytosol to the nucleus to promote the transcription of atrophy related genes, in particular two muscle-specific E3 ubiquitin ligases, the muscle RING finger-containing protein 1, MuRF1, and muscle atrophy F box protein, MAFbx. MuRF1 and MAFbx are both upregulated under several catabolic states and are extensively used as markers of muscle atrophy, while the deletion of either of them relieves skeletal muscle atrophy in mice [73,74]. On the other hand, FOXOs also activate autophagy through the transcriptional upregulation of autophagy-related factors in muscles during muscle denervation or starvation [75,76].

Akt is the common regulator of mTORC1 and FOXO signaling pathways. However, how the entire regulatory network responsible for muscle homeostasis is integrated remains to be explored. Several reports suggest that FOXO negatively suppress mTORC1. In muscles, FOXO1 downregulates mTOR, Raptor and other components of the mTOR signaling pathway [77]. In addition, 4EBP1, which is inhibited by mTORC1, is transcriptionally activated by FOXO in mouse skeletal muscle [77]. However, another study reports that expression of dominant negative FOXO3 suppresses fasting-induced autophagy without affecting mTOR activity in muscles, suggesting that the effects of FOXO on autophagy occur independently of mTORC1 signaling [75]. Interestingly, when FOXO3 is activated, constitutive and starvation-induced autophagy is still blocked by the mTORC1-mediated inhibition of ULK1 [65]. In contrast, the inhibition of mTORC1 activity by the deletion of Raptor induces autophagy, even though FOXO3 is suppressed and the FOXO3-dependent transcription of autophagy genes is decreased [65]. This study suggests that mTORC1 is the dominant regulator of autophagy induction in skeletal muscle. Although some studies indicate that FOXO and mTORC1 may function separately, more investigations are needed to further examine the crosstalk between FOXO, mTORC1, and autophagy (Figure 3).

## 4. Roles of the mTORC1-Autophagy Pathway in Muscle Regeneration

Muscle growth, maintenance, and regeneration mainly rely on a population of muscle stem cells (MuSCs), also known as satellite cells, which reside between the sarcolemma and the basal lamina [78]. Upon stimulation (such as exercise or injury), quiescent MuSCs proliferate and differentiate into myogenic progenitor cells, ultimately fusing with preexisting myofibers or each other to repair and replace damaged myofibers [79] (Figure 4). This process is regulated by multiple pathways including IGF-1/Akt, mTORC1, and autophagy [80,81,82].

### 4.1. mTORC1 in Myogenesis

In mice, the conditional knockout of *mTOR* or *Raptor*, but not *Rictor*, in embryonic muscle progenitors affects muscle development and results in perinatal lethality [82]. In adult muscle progenitors, the depletion of *mTOR* or *Raptor* in adult MuSCs inhibits MuSCs activation, proliferation, and differentiation and impairs skeletal muscle regeneration during muscle injury [82,83]. In contrast, *Rictor* knockout in adult MuSCs did not have a strong effect in skeletal muscle regeneration [82]. Consistent with these results, Rapamycin treatment, which inhibits mTORC1 activity, damages the growth of regenerating myofibers and the formation of nascent myofibers [82]. Thus, these studies suggest the central roles of mTORC1 in muscle regeneration.

Interestingly, the depletion of either *S6K1* or *4EBP1*, two well-known direct targets of mTORC1, does not fully phenocopy the *mTOR* knockout in muscle progenitors in response to muscle injury. S6K1 is dispensable for the initial formation of nascent myofiber, but it is required for myotube fusion [84]. 4EBP1 depletion increases myofiber growth but fails to impact MuSCs activation [81]. Thus, these studies suggest that, besides S6K1 and 4EBP1, other effectors downstream of mTORC1 are involved in muscle regeneration. Indeed, a recent study has found that mTORC1 phosphorylates and activates Per-Arnt-Sim domain kinase (PASK). mTORC1-activated PASK phosphorylates Wdr5 to promote MuSCs differentiation into myoblasts [85]. MicroRNAs (miRNAs) are small non-coding RNAs that bind to the 3′-UTR of target mRNAs and function in gene silencing and translational suppression. Several muscle-enriched miRNAs, known as myo-miRNAs, modulate muscle growth, development, and maintenance [86]. mTORC1 activity has been shown to regulate micro-RNA-1 (miR-1) expression. mTORC1-increased miR-1 promotes myoblast differentiation and in turn enhances muscle regeneration via the HDAC4-follistatin axis [87]. In addition to miR-1, the expression of several myo-miRNAs, such as miR-133, miR-206, and miR-125b, are regulated by mTORC1 directly or indirectly, suggesting an additional regulatory axis that mTORC1 exerts in muscle regeneration [88].

### 4.2. Autophagy in Myogenesis

Recent studies have shown that maintenance of the quiescent state of MuSCs require a basal level of autophagy. This basal, constitutive autophagy is crucial for MuSCs to maintain their stemness [80]. The disruption of autophagy in *Atg7*-deficient MuSCs results in mitochondrial dysfunction, and accumulation of damaged organelles and proteins, inducing oxidative stress and senescence [80,89]. Similar to *Atg7*-deficient MuSCs, MuSCs in old mice exhibit dramatically reduced activity of autophagy. This aging-induced autophagy defect causes the regenerative capacity of MuSCs to decline and induces senescence in aging mice [80,89]. Importantly, the reactivation of autophagy can reverse senescence and restore MuSCs stemness [80]. These findings reveal that autophagy is a key regulator of MuSCs.

During muscle injury, the activation of MuSCs requires autophagy to rapidly change cellular compositions, including the elimination of proteins involved in maintaining the quiescent state and the generation of new proteins related to cell-cycle regulation and differentiation [80]. Other studies also show that autophagy can provide a temporary energy source to fuel the initiation of proliferation during MuSCs activation and differentiation as the meager cytoplasm and mitochondria of the MuSCs fail to generate enough energy to exit quiescence [90,91]. Together, these studies demonstrate that autophagy is required for both MuSC quiescence and activation in response to muscle injury.

Similar to the effects of mTORC1 on MuSCs, both deficient and excessive autophagy result in a pathological cascade and lead to muscular weakness and atrophy symptoms. Increased autophagy can reduce the proliferative capacity of MuSCs, which plays an important role in the early regeneration of damaged skeletal muscle in myotonic dystrophy type 1 (DM1). The inhibition of autophagy by the overexpression of muscle blind-like 1 protein that increases phosphorylation levels of the mTORC1 can reverse the defective proliferation of MuSCs in myotonic dystrophy [92]. Lysine is essential for skeletal muscle growth, and its deprivation significantly decreases MuSCs viability and protein synthesis, while increasing autophagy and apoptosis. This process is mediated by mTORC1 activity [93]. These findings highlight the importance of balance between mTORC1 and autophagy in regulating MuSCs activities. The inactivation or hyperactivation of either one of the pathways induces muscle atrophy.

## 5. Potential Treatment of Muscle Diseases by Targeting mTORC1 and Autophagy

As mTORC1 and autophagy play vital roles in skeletal muscles, the imbalance of the mTORC1-autophagy axis has constantly been observed in human muscle diseases. Thus, the restoration of balance between mTORC1 and autophagy is a promising strategy to manage the progress of muscle diseases. Here, we focus on cancer cachexia and sarcopenia, as the roles of mTORC1 and autophagy have been extensively examined in these two muscle diseases.

### 5.1. Cancer Cachexia

Cancer cachexia is the wasting syndrome observed in cancer patients, affecting ~80% of advanced cancer patients and accounts for 20% of cancer deaths. It is characterized by skeletal muscle and adipose tissue loss [94,95]. Multiple studies have linked mTORC1 and autophagy to muscle atrophy induced by cancer cachexia. Decreases in mTORC1 signaling have been observed in skeletal muscles of mice inoculated with Lewis lung carcinoma (LLC) or colon-26 carcinoma (C26) cells [96,97,98]. Accordingly, autophagy activation has been shown to contribute to muscle loss during cancer cachexia [99]. Consistent with these results, the gastrocnemius muscles of Apc^Min/+^ mice, a model of colorectal cancer that develops cachexia, also exhibit a progressive reduction in mTORC1 activity [72]. Interestingly, the genetic activation of Akt–mTORC1 signaling in cachectic mice has been shown to reverse the 15–20% loss in muscle mass and strength [98]. Treadmill exercise results in a restoration of mTORC1 activity and improves cachexia phenotypes [72]. Furthermore, a natural plant product, salidroside, alleviates cancer cachexia phenotypes induced by C26 or LLC cancer cells and restored mTORC1 levels in muscles [97]. These studies suggest that reduction of anabolic mTORC1 signalling in skeletal muscle contributes to the loss of muscle mass during cachexia and that treatment of mTORC1 activators can possibly relieve cachexia phenotypes.

However, these findings somewhat contradict a report that mTORC1 inhibition can reduce muscle atrophy [100]. A decrease in autophagy flux is observed in the mice bearing C26 colon cancer tumor as well as in colon cancer patients. The treatment of either rapamycin or AMPK activator (AICAR) or aerobic exercise can restore autophagy in muscles to prevent cancer cachexia [100]. This study shows that enhanced autophagy prevents muscle loss in response to cachexia. While broad pharmaceutical inhibitors might have limited informative value, future studies using the genetic mouse model with inducible and muscle-specific gene knockouts/knock-ins will be needed to further investigate the physiological roles of mTORC1 and autophagy in cancer cachexia [27,98].

Although the effects of the modulation of mTORC1-autophagy axis on cancer cachexia in patients remains to be explored, some parameters related to muscle loss have been reported in clinical trials. An early study found that twenty patients treated with rapalogs (mTOR inhibitors) as monotherapy for at least 6 months showed significant decreases in skeletal muscle area revealed by CT-scans [101]. It is worthy to note that an untreated control of patients was not included in this study, therefore, the presence of other regulating factors cannot be excluded [101]. A clinical trial found that patients with advanced pancreatic neuroendocrine tumours receiving the rapalog Everolimus, an oral inhibitor of mTOR, showed weight loss and reduced appetite [102]. Similarly, multiple phase III trials have shown that cancer patients treated with rapalogs exhibited decreases in weight [103] and appetites [104,105,106]. These results suggest that mTORC1 inhibition may worsen cachexia (Table 1).

### 5.2. Sarcopenia

Sarcopenia refers to the progressive loss of muscle mass, strength, and function during the aging process [107]. It is highly associated with multiple adverse outcomes, such as poor quality of life, increased risks of falls, disability, morbidity, and mortality, leading to dramatically enhanced healthcare costs [108,109]. Multiple studies have identified mTORC1 hyperactivation as a crucial cause of sarcopenia [66,110,111,112], although its mechanisms remain unclear. The expression of the IGF-1 receptor increases in aging muscles [113]. However, a reduction of circulating levels of IGF-1 is observed in sarcopenic patients [114,115]. An increase in the IGF-1 receptor could be a compensatory response to lower IGF-1 levels, indicating that other pathways may be involved in aging-induced mTOR hyperphosphorylation. Indeed, several factors, such as physical activity and genetic background, can modulate mTORC1 signaling at the transcriptional level via non-coding RNAs (ncRNAs). Differentially expressed ncRNAs that have been linked to mTORC1 signaling were identified in the skeletal muscle of physically inactive older adults with or without exercise training [116]. In particular, miR-29c and miR-145-5p have been reported to modulate mTORC1 [117,118]. However, their connections to mTOR hyperphosphorylation during aging have not been fully explored.

Importantly, treatment with a low dose of rapalog RAD001, which inhibits mTORC1, increases skeletal muscle mass in sarcopenic rats [110]. Consistently, the stimulation of autophagy by physical exercise is reported to be an efficient intervention approach to renovating muscle structures and improving the overall quality of cell organelles [119]. During exercise, the reactivation of autophagy can also reverse senescence and restore MuSCs stemness [80]. These results show that the downregulation of mTORC1 and the stimulation of autophagy may exert protective effects against sarcopenia. In addition to exercise, current strategies targeting mTORC1 and autophagy against sarcopenia include dietary interventions, calorie restriction, and hormone treatment. Calorie restriction decreases the phosphorylation levels of key proteins of mTORC1 signaling in skeletal muscles, providing anti-aging effects [120]. The supplementation of resveratrol, a plant compound that mimics calorie restriction, is found to improve aging-induced muscle atrophy, possibly through autophagy [121,122]. Spermidine is a natural polyamine that stimulates autophagy. The external supplementation of spermidine has been shown to mitigate muscle atrophy during aging [123] and increase MuSCs proliferation [124] through autophagy. The treatment of hormones, such as with testosterone therapy, has been shown to enhance muscle mass and performance in the elderly, partially through mTORC1 signaling [125] (Table 1). Together, these findings suggest that the restoration of balance between mTORC1 and autophagy is a promising therapeutic strategy to manage the progression of sarcopenia.

**Table 1 ijms-24-00297-t001:** Strategies and studies targeting cancer cachexia and sarcopenia.

Disease	Intervention	Model	Main Findings	Reference
**Cancer cachexia**	Salidroside	Mice with C26 and LLC	Decreased mTORC1 in muscle	Chen et al., 2016 [97]
		C2C12 treated with TNF-⍺	Decreased mTORC1 in myotubes	
	*Raptor* KO mice	Mice with LLC and C26	Decreased mTORC1 activity Increased autophagic flux	Geremia et al., 2022 [98]
	IL-6 receptor Ab	*Apc^Min/+^* mice	Decreased mTORC1 activity Increased AMPK activity	White et al., 2011 [72]
	Aerobic exercise Rapamycin	C26-bearing mice and colon carcinoma patients	Loss of body weight Decreased autophagic flux	Pigna et al., 2016 [100]
	Rapamycin	C2C12 cells treated with conditioned media from C26 cells	Induced myotube atrophy Blockage of autophagic flux	
	Everolimus or Temsirolimus (Rapalog)	20 patients with renal cell carcinoma or pancreatic neuroendocrine tumour	Decreased muscle quantity, muscle mass and skeletal muscle area.	Gyawali et al., 2016 [101]
	Everolimus	410 patients with advanced pancreatic neuroendocrine tumors	Loss of body weight and reduced appetite.	Yao et al., 2011 [102]
	Everolimus with endocrine therapy, aromatase inhibitor	724 female patients with ER-positive breast cancer	Decreased body weight and reduced appetite.	Baselga et al., 2012 [103]
	Everolimus	302 patients with advanced, nonfunctional, well-differentiated lung or gastrointestinal neuroendocrine tumors	Decreased appetite.	Yao et al., 2016 [104]
	Temsirolimus	162 patients with relapsed or refractory MCL		Hess et al., 2009 [105]
	Nivolumab Everolimus	821 patients with advanced clear-cell renal-cell carcinoma		Motzer et al., 2015 [106]
**Sarcopenia**	Rapamycin	Male mice expressing LC3-GFP	Reactivation of autophagy prevented satellite cells senescence.	Garcia-Prat et al., 2016 [80]
	Rapalog RAD001	Old Male Sprague-Dawley rats (27 months)	Increased mTORC1 activation	Joseph et al., 2019 [110]
	Exercise	21-month-old Wistar rats	Increased autophagy via downregulating mTORC1 pathway.	Zeng et al., 2020 [119]
	Calorie restriction	Middle-aged Male Sprague-Dawley rats (20 months old)	Decreased mTORC1 activity	Chen et al., 2019 [120]
	Calorie restriction and resveratrol	Male FBN 25-month-old rats	Enhanced autophagy by inhibition of mTORC1 pathway	Dutta et al., 2014 [121]
	Calorie restriction and dietary resveratrol	Middle age to old age mice (14 months old to 30 months old)	Both dietary interventions prevent aging process through mTORC1	Barger et al., 2008 [122]
	Spermidine and exercise	D-galactose-induced aging rats with skeletal muscle atrophy	Increased autophagy	Fan et al., 2017 [123]
	Spermidine	8-week-old specific pathogen-free male C57/BL mice	Induced muscle atrophy and enhanced autophagy in MuSCs.	Zhang et al., 2018 [124]
	Testosterone therapy and resistance exercise	18 aged men (Ages 65–75 years)	Improve aging muscle performance and muscle mass via mTORC1	Gharahdaghi et al., 2019 [125]

## 6. Conclusions and Future Perspectives

As dysregulation of the mTORC1-autophagy axis has constantly been linked to human muscle diseases, the targeting of mTORC1 and autophagy is a promising strategy to treat muscle diseases. Although our understanding of the mTORC1-autophagy axis has been greatly advanced at the cellular level, how this signaling pathway is integrated or dysregulated at the physiological and pathological level remains largely unknown. For instance, to what extent can autophagy explain and rescue the effects induced by mTORC1 dysregulation in muscle diseases such as cancer cachexia or sarcopenia? Furthermore, as autophagy is required to provide energy for MuSCs activation, several questions remain as to which autophagic substrates are involved, which of these substrates are necessary, as well as how these substrates are altered in response to disease conditions or exercise. Would it then be possible to combine different activators or inhibitors of mTORC1/ autophagy to regain muscle homeostasis in muscle diseases? Animal genetics along with cell models will provide opportunities to solve these knowledge gaps. The major challenges that remain for the future are to translate this knowledge into pharmacological targeting of mTORC1 and autophagy to treat muscle diseases.

## Figures and Tables

**Figure 1 ijms-24-00297-f001:**
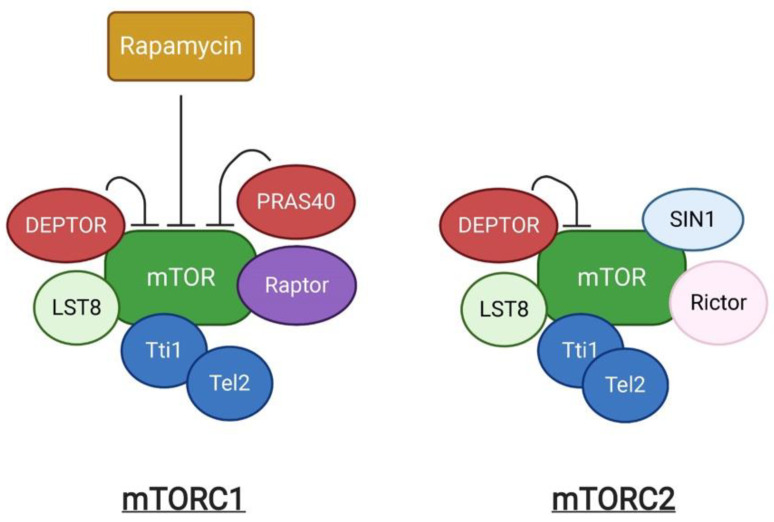
A schematic representation of mTOR complexes.

**Figure 2 ijms-24-00297-f002:**
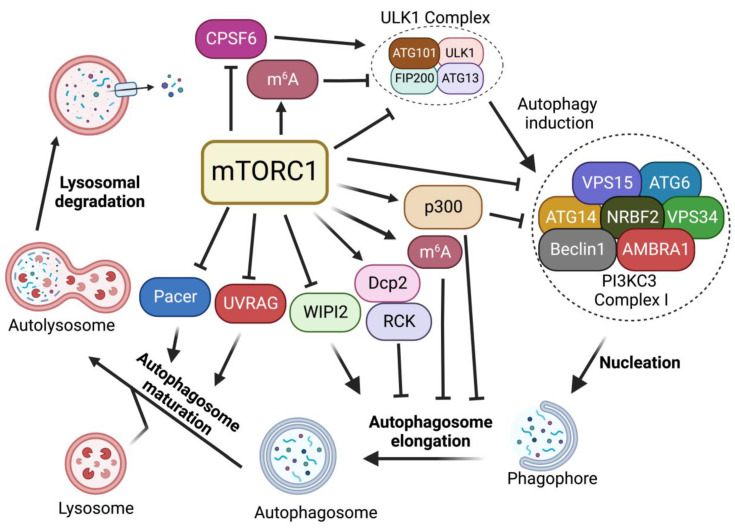
Regulation of autophagy by mTORC1. mTORC1 plays critical roles in the regulation of each step of the autophagy procedure, including autophagy initiation, nucleation, membrane expansion, and termination.

**Figure 3 ijms-24-00297-f003:**
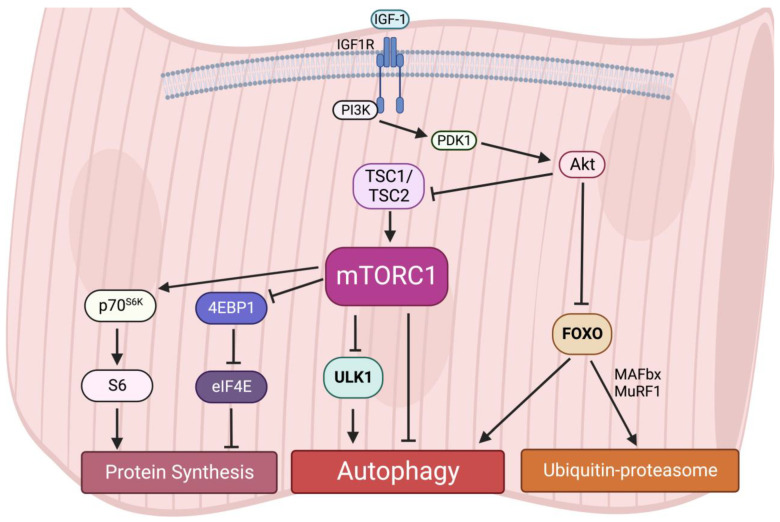
Protein homeostasis is regulated by the mTORC1-autophagy axis in muscles.

**Figure 4 ijms-24-00297-f004:**
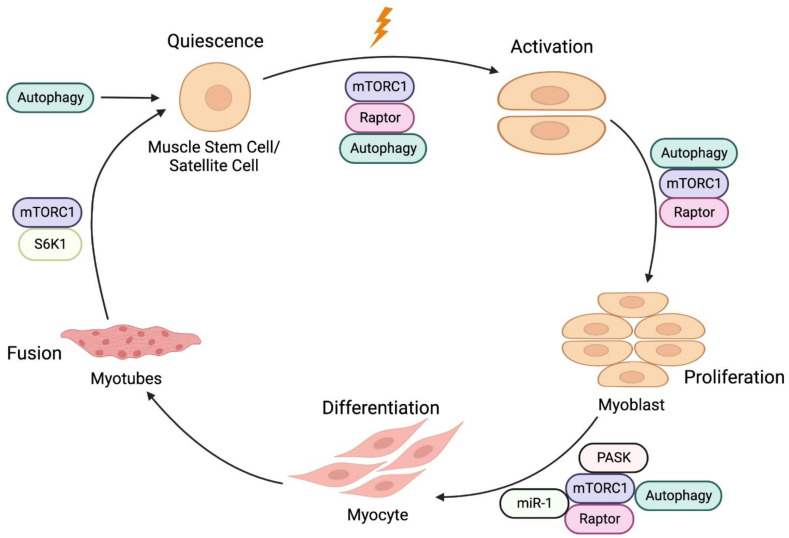
A schematic representation of muscle regeneration. Muscle stem cells (MuSCs) activation, proliferation, differentiation into myoblasts and fusion of myoblasts to form myofibers are under the regulation of mTORC1 signaling and autophagy.

## Data Availability

Not applicable.

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
