# Peer review of "The Importance of mTORC1-Autophagy Axis for Skeletal Muscle Diseases"

_ijms, 2022, doi:10.3390/ijms24010297_

Round 1

Reviewer 1 Report

The authors wrote a solid review on how mTORC1 regulates autophagy in skeletal muscle with an emphasis on cachexia. Figures are well designed and the basic medchanisms and key literature is covered.

Minor points:

1. The authors miss to mention that PKG via TSC2 is also able to regulate autophagy: e.g. in 2.1 "Regulation of ULK1 complex during autophagy initiation" / line 88 I suggest to mention that mTORC1-mediated autophagy via Ulk1 (at least in the heart) can also be regulated by protein kinase G (PKG) (Ranek et al. Nature 2019, and Oeing et al. Circ Res. 2020 and others).

This can also be commented on in paragraph 3/ line 229 as the cardiac phenotype is simliar with cardiac hypertrophy and dysfunctional autophagy (same references as above).

2. In Figure 3 I suggest to add Ulk1 between mTORC1 and autophagy as it is the key kinase to mediate autophagy by mTORC1.

3. Fig.4 figure legend should explain MuSC abbreviation.

4. it could be interesting to discuss that mTORC1-mediated autophagy is likely closely linked to cell metabolism (Düvel et al. Mol Cell 2010, or Kass lab Circ Res 2021).

5. it might be worth discussing that contradicting data (e.g. as mentioned in line 382) genetic models with complete knock-outs, transgenic (massive) overexpression or broad pharmaceutical inhibitors might have a limited informative value. A novel knock-in model like Ranek et al. Nature 2019 might be better to investigate the physiological role of mTORC1.

6. consider shortening the title to, e.g. "Balancing mTORC1 signaling and autophagy to treat skeletal muscle diseases"

7. mentioning other roles of mTOR: e.g. Ben-Sahra et al 2013 Science - pyrimidin synthesis, or metabolism (see above), translation, gene expression, ... 

English suggestions:

line 58, "in turn, it ..."

line 61 "increases activity of [4E-BP1]" - words are flipped

line 144 "the lysosome ... to degrade engulfed material"

Author Response

MS ID#: Revision of ijms-2088426

MS TITLE: The importance of mTORC1-autophagy axis for skeletal muscle diseases

We appreciate the thoughtful and constructive comments from both reviewers. Below is our detailed response to each of the comments made by the reviewers.

Reviewer #1

  1. The authors miss to mention that PKG via TSC2 is also able to regulate autophagy: e.g. in 2.1 "Regulation of ULK1 complex during autophagy initiation" / line 88 I suggest to mention that mTORC1-mediated autophagy via Ulk1 (at least in the heart) can also be regulated by protein kinase G (PKG) (Ranek et al. Nature 2019, and Oeing et al. Circ Res. 2020 and others).

We appreciate this reviewer for his/her constructive suggestions. We have included the roles of PKG1 in regulating mTORC1-dependent autophagy in section 2.1.

  1. In Figure 3 I suggest to add Ulk1 between mTORC1 and autophagy as it is the key kinase to mediate autophagy by mTORC1.

We have included ULK1 in Fig. 3.

  1. 3. Fig.4 figure legend should explain MuSC abbreviation.

We have revised Fig.4 figure legend accordingly.

  1. it could be interesting to discuss that mTORC1-mediated autophagy is likely closely linked to cell metabolism (Düvel et al. Mol Cell 2010, or Kass lab Circ Res 2021).

We appreciate this reviewer for his/her suggestions. We have included a paragraph briefly describing the link between autophagy and metabolism (Please see Section 3.2.).

  1. it might be worth discussing that contradicting data (e.g. as mentioned in line 382) genetic models with complete knock-outs, transgenic (massive) overexpression or broad pharmaceutical inhibitors might have a limited informative value. A novel knock-in model like Ranek et al. Nature 2019 might be better to investigate the physiological role of mTORC1.

We thank this reviewer for his/her suggestions. We have included a paragraph discussing the contradicting data (Section 5.1).

  1. consider shortening the title to, e.g. "Balancing mTORC1 signaling and autophagy to treat skeletal muscle diseases"

We have changed our title following the suggestion from reviewer 2.

  1. mentioning other roles of mTOR: e.g. Ben-Sahra et al 2013 Science - pyrimidin synthesis, or metabolism (see above), translation, gene expression, ... 

We thank this reviewer for his/her suggestions. mTORC1 indeed also play critical roles in diverse physiological functions, such as nucleotide synthesis or gene expression. In this review, we focused on the regulation of autophagy by mTORC1. Thus, we feel that these additional parts of mTORC1 functions are beyond the scope of the current review.

  1. English suggestions: line 58, "in turn, it ..."; line 61 "increases activity of [4E-BP1]" - words are flipped; line 144 "the lysosome ... to degrade engulfed material"

We have corrected the sentences following the reviewer’s suggestions.

Reviewer 2 Report

This is a comprehensive review on the role of mTORC1 in signaling and autophagy in general, and also in muscle. tissue.

However, the title of this review is somewhat misleading, since there is no dealing really with muscle diseases, instead the authors review the role of this molecule in the homeostasis of muscle tissue  and only in sarcopenia, which is a general state of muscle wasting occurring during the aging process.  From the title of the review one would expect in Section 5 at least a broader discussion on potential treatments, even theoretical, for  specific muscle diseases where autophagy have already been somewhat recognized. 

Dealing with specific disorders  would greatly improve the review.

Author Response

MS ID#: Revision of ijms-2088426

MS TITLE: The importance of mTORC1-autophagy axis for skeletal muscle diseases

We appreciate the thoughtful and constructive comments from both reviewers. Below is our detailed response to each of the comments made by the reviewers.

Reviewer #2

The title of this review is somewhat misleading, since there is no dealing really with muscle diseases, instead the authors review the role of this molecule in the homeostasis of muscle tissue and only in sarcopenia, which is a general state of muscle wasting occurring during the aging process.  From the title of the review, one would expect in Section 5 at least a broader discussion on potential treatments, even theoretical, for specific muscle diseases where autophagy has already been somewhat recognized. Dealing with specific disorders would greatly improve the review.

We thank this reviewer for his/her constructive suggestions. To avoid misleading readers, we have changed the title to “The importance of mTORC1-autophagy axis for skeletal muscle diseases”. We hope that this new title can convey our main idea appropriately.